# Novel Molecular Therapies and Genetic Landscape in Selected Rare Diseases with Hematologic Manifestations: A Review of the Literature

**DOI:** 10.3390/cells12030449

**Published:** 2023-01-30

**Authors:** Gabriela Ręka, Martyna Stefaniak, Monika Lejman

**Affiliations:** 1Independent Laboratory of Genetic Diagnostics, Medical University of Lublin, A. Gębali 6, 20-093 Lublin, Poland; 2Student Scientific Society of Independent Laboratory of Genetic Diagnostics, Medical University of Lublin, A. Gębali 6, 20-093 Lublin, Poland

**Keywords:** genes, hematology, rare diseases, therapy

## Abstract

Rare diseases affect less than 1 in 2000 people and are characterized by a serious, chronic, and progressive course. Among the described diseases, a mutation in a single gene caused mastocytosis, thrombotic thrombocytopenic purpura, Gaucher disease, and paroxysmal nocturnal hemoglobinuria (*KIT*, *ADAMTS13*, *GBA1*, and *PIG-A* genes, respectively). In Castleman disease, improper *ETS1*, *PTPN6*, *TGFBR2*, *DNMT3A*, and *PDGFRB* genes cause the appearance of symptoms. In histiocytosis, several mutation variants are described: *BRAF*, *MAP2K1*, *MAP3K1*, *ARAF*, *ERBB3*, *NRAS*, *KRAS*, *PICK1*, *PIK3R2*, and *PIK3CA*. Genes like *HPLH1*, *PRF1*, *UNC13D*, *STX11*, *STXBP2*, *SH2D1A*, *BIRC4*, *ITK*, *CD27*, *MAGT1*, *LYST*, *AP3B1*, and *RAB27A* are possible reasons for hemophagocytic lymphohistiocytosis. Among novel molecular medicines, tyrosine kinase inhibitors, mTOR inhibitors, BRAF inhibitors, interleukin 1 or 6 receptor antagonists, monoclonal antibodies, and JAK inhibitors are examples of drugs expanding therapeutic possibilities. An explanation of the molecular basis of rare diseases might lead to a better understanding of the pathogenesis and prognosis of the disease and may allow for the development of new molecularly targeted therapies.

## 1. Introduction

Rare diseases affect a very small number of people compared to the general population—less than 1 in 2000 people. They are characterized by a serious, often chronic, and progressive course. Due to the low frequency of rare diseases, delays in diagnosis and lack of treatment are some of the biggest patients’ problems [1]. The first symptoms of rare diseases often appear in childhood but may also occur in adulthood. Precise attention should be paid to the genetic basis of rare diseases, which allows us to make an accurate diagnosis and properly treat patients. Nowadays, treatment of many rare diseases is possible due to novel targeted therapies, thanks to the development of medicine and the increased emphasis placed on an individual approach to the patient. Genetic causes and therapeutic methods of the following chosen diseases with hematologic manifestations were analyzed: mastocytosis, Castleman disease, histiocytosis, thrombotic thrombocytopenic purpura, Gaucher disease, hemophagocytic lymphohistiocytosis, and paroxysmal nocturnal hemoglobinuria. The study aims to summarize diagnostic methods for pediatric rare diseases in which hematologic abnormalities are included in the clinical picture.

## 2. Mastocytosis

Mastocytosis is an orphan disease, defined as a heterogeneous group of disorders with many variants. In mastocytosis, clonal mast cells accumulate in various tissues and organs, like the skin, bone marrow, spleen, liver, and lymph nodes [2,3,4]. Mediators from mast cells and the anatomical distribution of the cells are responsible for the symptoms of the disease [3]. The frequency of systemic mastocytosis is estimated at 1–5/10 000 [5]. There is no difference in children between the prevalence of mastocytosis among males and females, and no race is predominant; however, precise data are limited [4].

For cutaneous mastocytosis (CM), the main symptoms include skin lesions. For systemic mastocytosis (SM), symptoms include cytopenia due to massive infiltration of bone marrow, signs of anemia, hemorrhagic diathesis, susceptibility to infections, hepatomegaly, splenomegaly, impaired liver function with ascites and/or portal hypertension, malabsorption, weight loss, osteolytic bone lesions and/or pathological fractures, and even severe anaphylaxis [6,7]. Among children, the cutaneous manifestation of mastocytosis is most frequently seen, in about 85% of cases, and has a good prognosis [2,8,9]. Pediatric-onset mastocytosis is often diagnosed before 2 years of age, usually as urticaria pigmentosa [4]. Skin lesions are described as red to brown to yellow, 1-2 cm in diameter, macules, plaques, or nodules, mainly in the trunk and extremities. Stroking or rubbing can induce erythema, swelling, and blister formation, which relate to pruritus and dermatographism [4].

The diagnostic criteria for mastocytosis in children are based on studies among adults [8]. According to the World Health Organization diagnostic criteria for systemic mastocytosis from 2016, the diagnosis can be established when at least one major and one minor or three minor criteria are stated. Multifocal dense infiltrates of mast cells (≥15 mast cells in aggregates) in bone marrow biopsies and/or in sections of other extracutaneous organ(s) is the major criterion, whereas the minor criteria are as follows: the presence of atypical (type I or type II) or spindle-shaped morphology in >25% of all mast cells; detection of a *KIT* point mutation at codon 816 in the bone marrow or another extracutaneous organ; expression of CD2 and/or CD25 by the mast cells in bone marrow or blood or another extracutaneous organ; and a baseline serum tryptase level above 20 ng/mL (in case of an unrelated myeloid neoplasm, the final minor criterion is not valid as a systemic mastocytosis criterion) [6,7,8,9,10]. According to a recent study in adults, E-selectin, adrenomedullin, T-cell immunoglobulin, mucin domain 1, and CUB domain-containing protein 1 (CDCP1)/CD138 were other proteins elevated in mastocytosis. Allergin-1 and pregnancy-associated plasma protein-A (PAPP-A) were decreased in patients with anaphylaxis, whereas galectin-3 was increased [6].

Inheritance is not seen in most cases of mastocytosis. In familial cases, an autosomal dominant inheritance pattern with incomplete penetrance was noted [11]. A point mutation in codon 816 of the *KIT* gene (*KIT* D816V) is typically found in adults with systemic mastocytosis. The prognostic influence of the *KIT* gene mutation in codon 816 in pediatric mastocytosis is unknown [8]. *KIT* codes for c-kit, a membrane receptor for stem cell factor, which is expressed in the surface membrane of the mast cells. Monozygotic twins and triplets have been reported [11]. However, family cases are rarely seen, and in the familial and childhood variants, no single gene has been identified [4,12]. The single nucleotide polymorphism causing a Met-541-Leu c-kit mutation might predispose to mastocytosis among children [12]. Sporadic mutations in c-kit at codons 816 and 820 and inactivating mutations at codon 839 were described in 43% of pediatric patients with cutaneous mastocytosis (skin biopsies). Patients with the Asp816Phe mutation acquire the disease prior to patients with Asp816Val mutations. The missense activating mutations Asp816Val and Asp816Phe were noted in those with mastocytomas, urticaria pigmentosa, and diffuse cutaneous mastocytosis [4]. In children, the mutational pattern is distinct and more commonly involves the extracellular domains of *KIT* (exons 8 and 9) [10]. In the literature, the following mutations were reported: at codon 816 of *KIT* within the tyrosine kinase domain (D816Y, D816F, D816I, and D816G) and mutations at nearby codons (L799F, I817V, N819Y, D820G, N822I, N822L, InsVI815-816, E839K, S840N, and S849I) [10]. Van den Poel et al. noticed the *KIT* D816V mutation in patients with systemic mastocytosis (1 child and 37 adults) and basic tryptase elevation in 16 out of 19 patients diagnosed with SM [13]. Shibata et al. reported a case of a 4-month-old male with diffuse cutaneous mastocytosis in whom the presence of a deletion of codon 419 in exon 8 (c.1255_1257delGAC [p. Asp419del]) was detected [14]. Genetic profiling of *KIT* and characterization of associated gene mutations by next-generation sequencing (NGS) panels enable the division of patients into three prognostic subgroups: patients with multilineage *KIT* D816V involvement, patients with mast cell-restricted *KIT* D816V, and patients with “multi-mutated disease ” [7].

Therapy of mastocytosis is aimed at alleviation of symptoms [3]. Avoidance of triggering factors, leukotriene antagonists, H1 and H2 antihistamines, cromolyn sodium, corticosteroids, and methoxypsoralen therapy with long-wave psoralen plus ultraviolet A radiation (PUVA) can be mentioned as methods of treatment [4]. As c-kit mutations play a role in the etiology of the disease, targeted therapies using kit inhibitors might be promising treatment options [3]. A KIT tyrosine kinase inhibitor, midostaurin, which inhibits KIT D816V, has recently been introduced. Gotlib et al. evaluated in an open-label study the effectiveness of oral midostaurin at a dose of 100 mg twice per day in 116 patients with advanced systemic mastocytosis. Complete resolution of at least one type of mastocytosis-related organ damage was noted in 45% of patients. The overall response rate was described as 60% (95% confidence interval, 49 to 70) [15,16]. Imatinib, nilotinib, dasatinib, and masitinib exert favorable effects on mediator-related symptoms of mastocytosis. Novel inhibitors, such as avapritinib or ripretinib, are in clinical development [7]. Avapritinib, a KIT and PDGFRα (platelet-derived growth factor receptor A) inhibitor, was precisely projected to inhibit KIT D816V [16]. There is a possibility of using mTOR inhibitors like rapamycin for patients expressing D816V-mutated *KIT* in aggressive systemic mastocytosis [7]. Cladribine, which targets nucleoside metabolism, IFN-α, and allogeneic hematopoietic stem cell transplantation (HSCT) might be considered for patients with advanced mastocytosis [9]. Barete et al. assessed the efficacy and safety of cladribine in 68 adults with indolent or advanced mastocytosis in a scheme of 0.14 mg per kg in infusion or subcutaneously in days 1–5, repeated at 4–12 weeks until 1 to 9 courses. Cladribine (2-chlorodeoxyadenosine) is a synthetic purine analog cytoreductive medicine with a 72% overall response rate (complete remission [R]/major/partial R: 0%/47%/25%) [17].

## 3. Castleman Disease

Castleman disease (CD) is a heterogeneous non-malignant lymphoproliferative disease with an estimated incidence of 5 per million person-years [18]. The term describes a group of disorders that share a spectrum of characteristic histopathological features, including atrophic or hyperplastic germinal centers, prominent follicular dendritic cells (FDCs), hypervascularization, polyclonal lymphoproliferation, and/or polytypic plasmacytosis.

Castleman disease is divided into two types: unicentric Castleman disease (UCD) associated with single-node region adenopathy and multicentric Castleman disease (MCD) associated with multiple node region adenopathies. Unicentric Castleman disease and multicentric Castleman disease may be associated with Kaposi sarcoma, non-Hodgkin and Hodgkin lymphoma, and POEMS syndrome [19]. Multicentric Castleman disease is classified into idiopathic MCD (iMCD), human herpes virus-associated MCD-8 (HHV8-MCD), and polyneuropathy, organomegaly, endocrinopathy, monoclonal plasma cell disorder, skin changes (POEMS)-associated MCD (POEMS-MCD). iMCD can be further divided into iMCD-thrombocytopenia, ascites, reticulin fibrosis, renal dysfunction, organomegaly (iMCD-TAFRO), or iMCD-not otherwise specified (iMCD-NOS). The histopathological features of the various forms of Castleman disease vary and are mostly non-specific, as they are observed to a varying extent in different clinical variants and autoimmune and infectious diseases [20,21,22,23]. Castleman disease is rare in children, can be misdiagnosed because it has no specific manifestations, and the prognosis depends on the subtype [24]. Thrombocytopenia, ascites/anasarca, myelofibrosis/fever, renal dysfunction/reticulin fibrosis, and organomegaly (TAFRO) constitute a distinct clinicopathological form of idiopathic HHV8/KSHV-negative Castleman disease with mixed histological features of hypervascular and plasmacytoid lymphocytes [20,21,22,23].

The pathogenesis of UCD is most likely determined by a neoplastic follicular dendritic cell population. HHV-8-associated MCD pathogenesis is caused by viruses, whereas POEMS-MCD pathogenesis is committed to a monoclonal plasma cell population. iMCD is poorly understood, although clinical data suggest a pathologic role for interleukin-6 [25].

Histopathology is the key to the diagnosis of Castleman disease, which is usually classified into three types: the glassy vascular type (HV type), the plasma type (PC type), and the mixed type. The HV type is more common in patients with UCD and manifests mainly follicular dysplasia, degeneration of the germline center, widening of the mantle, atrophy of the lymphatic sinus, or fibrosis. Another feature is the intravesical growth of endothelial blood vessels in the area with the vitreous lesion; it can penetrate the germline center, giving the appearance of a "lollipop". The PC type is more common in patients with MCD. The pathology is mainly characterized by follicular growth and infiltration of plasma cells; there are few vitreous vessels and onion peels in the interalveolar area, and the lymphatic sinuses are preserved. There are fewer hybrid types, and both have the above characteristics [19]. Assessment of Castleman disease should include, in addition to histological evaluation with immunological staining, a series of laboratory and radiological examinations, and PET system imaging, which can provide information on the metabolic activity of the affected lymph nodes and help determine the severity of the disease. Recommended laboratory tests include screening for anemia, elevated CRP and/or erythrocyte sedimentation rate (ESR), hypoalbuminemia, hypergammaglobulinemia, and other markers of cytokine-induced inflammation. It should be noted that most cases of UCD are asymptomatic, and often there are no laboratory abnormalities [26,27]. High levels of IL-6 that cannot be otherwise explained may be one of the potential diagnostic criteria for MCD. Most clinical features and laboratory abnormalities in patients with MCD are associated with IL-6 overexpression, and basic information on the differential diagnostic criteria for MCD can be identified based on patients’ clinical and pathological characteristics. Although the international evidence-based consensus diagnostic criteria for HHV-8 negative/idiopathic MCD were published in 2017, more studies are needed to define the criteria for the diagnosis of MCD due to the lack of extensive epidemiological data [28].

Castleman disease is not considered to be inherited, and it occurs sporadically in people without a family history. Nagy A. et al. analyzed 15 cases of UCD and 3 cases of iMCD using next-generation directed sequencing (NGS; 405 genes), as well as 3 cases of FDCS associated with the vitreous vascular variant UCD (UCD-HVV) using whole exome sequencing. Typical amplification of *ETS1, PTPN6,* and *TGFBR2* as observed in one case of iMCD and one case of UCD. The iMCD case also had the somatic *DNMT3A* L295Q mutation. This iMCD patient also exhibited clinicopathological features corresponding to a specific subtype known as Castleman-Kojima disease (thrombocytopenia, anasarca, fever, reticulin fibrosis, and the clinical subtype of organomegaly [TAFRO]). In addition, one case of UCD-HVV showed amplification of a histone gene cluster on chromosome 6p. UCD-HVV-associated FDCS have demonstrated mutations and copy number alterations in known oncogenes, tumor suppressors, and chromatin remodeling proteins [29]. In another study, recurrent *PDGFRB* mutations encoding p.Asn666Ser were detected in patients with UCD, which strongly suggests that *PDGFRB* mutations in stromal cells may play a key role in the pathogenesis of UCD [30]. There is a need for in vivo functional studies to determine how particular genetic changes affect the phenotypic symptoms of UCD and iMCD and to fully study all genes, intron regions, and translocations. 

The advent of effective antiretroviral therapy and the use of rituximab improved the results in the treatment of HHV8-MCD. Therapies targeting interleukin 6 (like tocilizumab) are highly effective in many iMCD patients, but other therapies (such as corticosteroids, rituximab, thalidomide, lenalidomide, bortezomib, cyclosporine, sirolimus, or interferon) are required in refractory cases [18]. The subtypes of Castleman’s disease are presented in Figure 1.

## 4. Langerhans-cell Histiocytosis

Langerhans-cell histiocytosis (LCH), the most common histiocytic disorder, encompasses conditions characterized by aberrant function and differentiation or proliferation of cells of the mononuclear phagocyte system. It is diagnosed approximately in 1–9/10,000 people [5]. Childhood LCH ranges from 3.5 to 7 cases per 1,000,000 children annually [31]. LCH has a widely variable clinical presentation, ranging from single indolent lesions to explosive multisystem disease. Bone, skin, pituitary gland, lung, central nervous system, and lymphoid organs are the main organs involved, whereas liver and intestinal tract localizations are less frequently encountered. Children with lesions in the liver, spleen, or bone marrow are classified as having high-risk LCH due to being at the highest risk for death [32,33].

LCH is caused by the clonal expansion of myeloid precursors that differentiate into CD1a+/CD207+ cells in lesions, which leads to a spectrum of organ involvement and dysfunction. Studies have shown that LCH cells originate from myeloid dendritic cells rather than skin Langerhans cells. The pathogenic cells are defined by constitutive activation of the MAPK signaling pathway [34,35].

LCH is generally considered a non-hereditary, sporadic disease. Since LCH may affect any organ or system of the body, the condition should be considered whenever suggestive clinical manifestations occur in the skin, bone, lung, liver, or central nervous system (CNS). A definitive diagnosis of LCH requires a combination of clinical presentation, histology, and immunohistochemistry. The inflammatory infiltrate contains various proportions of LCH cells, the disease hallmark, which are round and have characteristic “coffee-bean” cleaved nuclei and eosinophilic cytoplasm. Positive immunohistochemistry staining for CD1a and CD207 (langerin) is required for a definitive diagnosis [36,37]. A gain-of-function mutation in *BRAF* (V600E) was identified in more than half of LCH patient samples in research from 2010 [35]. A somatic mutation of *BRAF* that causes the alteration of RAS-RAF-MEK-ERK cell signaling pathway is the most common genetic abnormality associated with LCH and is a poor prognostic marker [38]. Mutations of *MAP2K1, MAP3K1, ARAF, ERBB3, NRAS, KRAS, PICK1, PIK3R2,* and *PIK3CA* were also described in the literature as a cause of the condition [32,34,35]. Smoking is the sole known risk factor, but a significant effect of smoking cessation on the course of disease could not be confirmed [39]. There is a need for a high index of suspicion for the diagnosis of LCH due to frequent misdiagnosis. In addition to survival data and the analysis of prognostic factors, the prospective collection of data on diverse presentations is essential [40].

Treatment of LCH is risk-adapted; patients with single lesions may respond well to local treatment, whereas patients with multi-system disease and risk-organ involvement require more intensive therapy. Treatment with BRAF inhibitors, such as vemurafenib and dabrafenib, has been shown to induce complete and durable responses, and the role of BRAF and MEK inhibitors is currently being investigated [35,41]. Optimal therapy for patients with single-system bone LCH has not been established. Less toxic therapeutic approaches should be considered for these patients [42]. Among targeted therapies, imatinib, a tyrosine kinase inhibitor that targets the receptors expressed in LCH, has shown efficacy in patients with refractory multisystem LCH [35].

## 5. Thrombotic Thrombocytopenic Purpura

Thrombotic thrombocytopenic purpura (TTP) is a rare condition in which severe thrombocytopenia, microangiopathic hemolytic anemia, and microvascular blood clots rich in platelets might lead to ischemic injury of an end organ [43]. Thrombotic thrombocytopenic purpura can be immune-mediated/acquired (iTTP) or congenital (cTTP). Congenital TTP is also known as Upshaw-Schulman syndrome [43,44]. The incidence of iTTP is estimated at 1/165,000–1,000,000 and that of cTTP ranges from 1/60,000–2,500,000 [5]. Immune TTP accounts for approximately 95% of all thrombotic thrombocytopenic purpura [45].

Immune TTP is a life-threatening blood disorder, the clinical features of which include severe thrombocytopenia, microangiopathic hemolytic anemia, fever, and renal and neurologic dysfunction. Ischemic stroke, renal insufficiency, and myocardial ischemia might be consequences of end-organ damage. Analysis of peripheral blood shows low hemoglobin and hematocrit, low haptoglobin, elevated serum lactate dehydrogenase, and the presence of schistocytes [43,45,46,47]. In 10% of all immune TTP cases, symptoms present in childhood [43]. Congenital TTP presents as episodic microangiopathic hemolytic anemia, thrombocytopenia, and damage to internal organs. The disease might be diagnosed in neonates, and it can also present for the first time in adults [48]. Toret et al. described a case of cTTP in a 12-year-old boy. The patient presented with jaundice and a skin rash. Blood analysis revealed nonimmune hemolytic anemia, severe thrombocytopenia, 8% schistocytes, polychromasia, and anisocytosis [49].

Immune-mediated TTS is a result of anti-ADAMTS13 (a disintegrin and metalloprotease with thrombospondin type 1 repeats, member 13) autoantibodies and a severe deficiency of ADAMTS13. Congenital TTP is a consequence of biallelic mutations in the *ADAMTS13* gene [43,50]. Modifying factors such as sex, ethnicity, and obesity, as well as genetic risk factors for autoimmunity at the human leukocyte antigen class II locus DRB1*11 and DQB1*03 alleles and the protective allele DRB1*04, are involved in the loss of tolerance towards *ADAMTS13* [51].

Congenital TTP is inherited in an autosomal recessive manner. Nonaka et al. described a case of a family with cTTP in which the patient’s parents were heterozygous carriers of *ADAMTS13* mutations (p.R193W, c.577C>T, exon 6 in the father, and p.H1141Tfs*85, c.3421del, exon 25 in the mother, and no ADAMTS13 mutation in her brother). Therefore, the patient was a compound heterozygote of p.R193W (c.577C>T, exon 6) and p.H1141Tfs*85 (c.3421del, exon 25) [44]. In a case report by Toret et al., the DNA sequence analyses showed compound heterozygosity consisting of c.291_391del in exon 3 and c.4143dupA in exon 29 in a 12-year-old boy with cTTP [49]. Wang and Zhao described a case of a neonate with a novel variant of a missense compound heterozygous mutation in *ADAMTS13*, c.1187G>A/c.1595G>T. High-throughput sequencing, polymerase chain reaction, and Sanger sequencing were used in genetic screen testing. It was reported that *ADAMTS13* mutation analysis was only performed in 8 of the 12 cases of congenital TTP in neonates that have been reported globally [46].

Therapeutic plasma exchange with fresh frozen plasma replacement is given as the front-line therapy for TTP. Immunosuppressive therapy with glucocorticoids, cyclosporine A, or mycophenolate mofetil has shown efficacy [43]. Measurements of ADAMTS13 activity have become, in clinical practice, not only diagnostic markers but also an indicator of recurrence and response to therapy [52]. Knowledge of the molecular cause of the disease allowed the off-label use of rituximab in iTTP. Rituximab is an anti-CD20 monoclonal antibody that suppresses anti-ADAMTS13 autoantibodies [50]. Safety and effectiveness of rituximab were evaluated in 22 adults in an open-label prospective study by Froissart et al. Patients with severe, acquired TTP who responded poorly to therapeutic plasma exchange and who were treated with add-on rituximab therapy (four infusions over 15 days) presented with reduced overall treatment duration and shorter 1-year relapses than controls [53]. Bortezomib, which is a proteasome inhibitor targeting plasma cells, appears to be effective as an alternative to rituximab. Caplacizumab, a humanized immunoglobulin that targets the A1 domain of von Willebrand factor, prevents its interaction with platelets, blocks platelet aggregation, and reduces time to platelet count normalization [43]. Peyvandi et al., in the second phase of a randomized controlled study, observed the effectiveness of caplacizumab in 75 patients with acquired TTP (36 received caplacizumab and 39 received a placebo). The time to a response was significantly reduced (39% reduction, *p* = 0.005) with caplacizumab as compared with placebo [54].

## 6. Gaucher Disease

Gaucher disease (GD) is a rare genetic disease caused by a deficiency of the lysosomal enzyme glucocerebrosidase that leads to the accumulation of its substrate, glucosylceramide, in macrophages. In the general population, its incidence varies between 0.4 and 5.8/100,000 inhabitants [55].

Type 1 Gaucher disease affects most patients and is characterized by its huge heterogeneity, including asymptomatic forms and more severe presentations. The most frequent symptoms are anemia, thrombocytopenia, splenomegaly, and/or hepatomegaly, as well as potentially severe bone involvement with avascular osteonecrosis (AVN), osteoporosis, fractures, and lytic lesions. This type is associated with a higher risk of some solid cancers, Parkinson disease, and hematologic diseases, particularly multiple myeloma. Type 2 and type 3 Gaucher diseases are associated with neurological involvement, either severe in type 2 or variable in type 3 [55,56,57].

GD may come to light because of investigations for visceromegaly or pancytopenia. Therefore, Gaucher cells may be identified on tissue biopsy specimens, principally those of the bone marrow (during investigations for splenomegaly or cytopenias) or liver (during investigations for hepatomegaly or abnormal liver-related biochemical tests). However, specific diagnosis is made by measuring acid β-glucosidase activity in fresh peripheral blood leukocytes or occasionally by enzymatic analysis of fibroblasts cultured from skin biopsy specimens. Confirmation and better characterization of the condition may subsequently be afforded by the identification of biallelic pathogenic variants in glucocerebrosidase gene (*GBA1*), which encodes lysosomal GBA [57,58,59]. MRI is useful for monitoring skeletal involvement because it provides a semi-quantitative assessment of marrow infiltration and the degree of bone infarction [60].

Gaucher disease is inherited in an autosomal recessive manner. Newly available techniques in molecular biology enabled the characterization of the *GBA1*. The gene was localized to chromosome 1q21 by in-situ hybridization analysis. The *GBA1* cDNA served as a probe to identify and isolate clones from controls and patients. The gene was found to encompass 11 exons spanning around 7000 base pairs. Almost immediately, it was recognized that a highly homologous pseudogene was present near *GBA1*. The elucidation of the full sequence of *GBA1* ultimately enabled the production of recombinant proteins for therapeutic use. The first mutation in the *GBA1* identified was a C to T substitution in exon 10, resulting in the replacement of a proline for leucine at amino acid position 444 [L483P]. Identification of the common N370S [N409S] mutation was later found in a patient with type 1 Gaucher disease. To date, more than 300 different *GBA1* mutations have been described. The mutation nomenclature is at times confusing, as the numbering of the affected amino acids was eventually changed to include the 39 amino acid leader sequence [61].

Specific treatment, such as enzyme replacement therapy (ERT) using one of the currently available molecules such as imiglucerase, velaglucerase, or taliglucerase, or substrate reduction therapy, is indicated in symptomatic type 1 Gaucher disease. Only ERT is indicated in type 3 Gaucher disease. The approval of ERT for GD in the pediatric age group has significantly altered the course of the disease, especially for non-neuronopathic and chronic neuronopathic forms, as ERT does not cross the blood-brain barrier. Treatment improves the quality of life and prognosis. The rarity of Gaucher disease and its wide variability in clinical presentations lead to diagnosis delays [55,56,62]. Miglustat or eliglustat are inhibitors of the biosynthesis of glucosylceramide that are possible to use in Gaucher disease [55].

## 7. Hemophagocytic Lymphohistiocytosis

Hemophagocytic lymphohistiocytosis (HLH), also known as hemophagocytic syndrome, is caused by overactivated macrophages and histiocytes that result in excessive cytokine release, destruction of hematopoietic cells, and multiorgan dysfunction [63,64]. HLH is a rare disease affecting mainly children but also adults. The course of HLH is life-threatening unless effective treatment is instituted [65]. The prevalence is 1.2/1 million/year in children. In adults, the disease is diagnosed less frequently [66].

The etiology of HLH is different in the adult and pediatric populations. Although there is no single specific and sensitive diagnostic test for HLH, various clinical and laboratory findings should be taken into consideration. Patients with an HLH-associated gene defect and/or at least five of the following eight criteria can be diagnosed with HLH: fever, low or absent natural killer cell function, cytopenias, splenomegaly, increased triglycerides or low fibrinogen, high ferritin, hemophagocytosis, and elevated soluble CD25 (interleukin 2 receptor alpha (IL2Rα)) [67].

HLH was primarily considered to be only a genetic disorder; however, secondary HLH can be triggered by infections, malignancies, autoinflammatory, and rheumatologic disorders. Familial HLH is caused by mutations at specific gene loci (*HPLH1, PRF1, UNC13D, STX11*, and *STXBP2*), which code for proteins with a fundamental role in lymphocyte cytotoxicity [67,68]. Mutations in the *HPLH1* gene are responsible for familial hemophagocytic lymphohistiocytosis type 1 (FLH-1), a mutation in the *PRF1* gene causes FLH-2, mutations in the *UNC13D* gene cause FHL-3, mutations in the *STX11* gene cause FHL-4, and a mutation in the *STXBP2* (*UNC18B*) gene causes FHL-5 [69]. HLH and lymphoproliferative disease can be caused by mutations in the following genes: *SH2D1A, BIRC4, ITK, CD27*, and *MAGT1*. They encode signaling proteins that play a role in the activation, survival, differentiation, and migration of NK and T cells [70]. Chediak-Higashi syndrome (mutations in *LYST*), Hermansky-Pudlak syndrome type 2 (mutations in *AP3B1*), and Griscelli syndrome (mutations in *RAB27A*) are immunodeficiencies with high rates of developing HLH [61]. Familial hemophagocytic lymphohistiocytosis results from a distinct set of autosomal-recessive gene mutations of lymphocyte cytotoxicity [64]. Shabrish et al., in their study of 101 Indian patients, found that 53% patients harboring homozygous mutations presented at the median age of 10 months, and patients with compound heterozygous mutations had onset of disease at the median age of three. Twelve patients with a monoallelic mutation in FHL genes had first symptoms of disease at a median age of 10 months [68]. GATA2 deficiency was described in the literature in patients with acute secondary HLH [64]. Lam et al. found a de novo CDC42 mutation (Chr1:22417990C>T, p.R186C) in four unrelated patients with NOCARH syndrome (neonatal-onset cytopenia with dyshematopoiesis, rash, autoinflammation, and HLH) [63].

Treatment of HLH includes immunosuppressive drugs, such as corticosteroids, etoposide, and cyclosporin [65,67]. Recently, novel molecular-targeting drugs have emerged. Emapalumab, which is a human anti-IFN-γ monoclonal antibody, was registered for the treatment of patients with refractory HLH [67]. Locatelli et al., in an open-label, single-group, phase 2–3 study, assessed the efficacy and safety of emapalumab administered with dexamethasone in HLH in 34 patients at the age of 18 or younger (27 who had received conventional therapy before enrollment and 7 who had not). A response was noted among 63% of the previously treated patients and 65% of the patients who received an emapalumab infusion [71]. Anakinra (interleukin 1 receptor antagonist) and tocilizumab (interleukin 6 receptor antagonist), which block cytokines, and ruxolitinib, tofacitnib, baricitinib, and itacitinib, which are JAK inhibitors, can be mentioned as examples of molecular drugs expanding therapeutic possibilities [67]. Treatment with ruxolitinib as monotherapy or combination therapy (in upfront and salvage settings) showed fast, sustained improvement in clinical status, hematological cell counts, and inflammatory markers followed by persistent remission among 4 patients with profound secondary HLH [72].

## 8. Paroxysmal Nocturnal Hemoglobinuria

Paroxysmal nocturnal hemoglobinuria (PNH) is an infrequent intravascular hemolytic anemia in which hemolysis occurs by the complement system [73]. It is a chronic, progressive, multi-systemic, and life-threatening disease that results from the expansion of a clone of hematopoietic cells [74]. Its prevalence is stated as 1–9/100,000 [5].

Symptoms include a classic triad of hemolytic anemia, thrombosis, and failure of the bone marrow. PNH is a rare condition in children (5–10% of cases). However, it should be taken into consideration in the differential diagnosis, particularly in children with acute kidney injury. Common symptoms in children include pallor, fatigue, weakness, hemorrhage, thrombosis, and isolated hemoglobinuria [75].

The gold standard test to confirm PNH is flow cytometry performed on peripheral blood that detects very small PNH clones (<1% of a patient’s hematopoiesis) [74,76,77]. PNH can be caused by an acquired mutation in the phosphatidylinositol-N-acetylglucosaminyltransferase-subunit-A gene (*PIG*-A) that leads to the deficiency of cellular anchors for complement inhibitor proteins cluster of differentiation CD55 (decay accelerating factor, DAF, which stabilizes C3 and C5 convertase) and CD59 (membrane inhibitor of reactive lysis, MIRL, which inhibits membrane attack complex formation) [70,75]. *PIG*-A is located on the X chromosome (Xp22.1) [5,75]. CD55 and CD59 inhibit complement activation and prevent healthy cells from undergoing complement-mediated lysis [75]. A lack of them leads to suboptimal complement inhibition and complement-mediated hemolysis of erythrocytes [76]. Jeong et al. found a strong positive correlation between paroxysmal nocturnal hemoglobinuria clone size by flow cytometry and variant allele frequency mutations of the *PIG* gene [76]. Recently, the complement inhibitor eculizumab, a monoclonal antibody targeting C5, has been introduced. It significantly reduces hemolysis, anemia, the occurrence of thrombosis, and morbidity and mortality [76].

Treatment of PNH includes anti-thrombosis prophylaxis, blood transfusions, and allogeneic bone marrow transplantation. Recently, the complement inhibitor eculizumab, a monoclonal antibody targeting the protein C5, has been introduced. It significantly reduces hemolysis, anemia, the occurrence of thrombosis, morbidity, and mortality [74,76]. In a meta-analysis of six studies by Zhou et al., a total of 235 patients treated with eculizumab were included. Eculizumab was safe and effective at decreasing lactate dehydrogenase (LDH) levels and transfusion rates while increasing hemoglobin levels [78]. In 2021, pegcetacoplan was approved to treat adults with PNH. It is a pegylated pentadecapeptide that targets complement C3 to control intravascular and extravascular hemolysis. The study by Hillmen et al. indicated that pegcetacoplan was superior to eculizumab in clinical and hematologic outcomes in PNH patients [79]. Another well-tolerated drug was ravulizumab. Outcomes from a third phase of a randomized trial of ravulizumab in adults with PNH showed that those on stable eculizumab therapy who received ravulizumab over 52 weeks presented with durable efficacy. Further efficacy was noted in adults who received eculizumab during the primary evaluation period and then changed treatment to ravulizumab [80].

A summary of the genetic landscape and methods of modern treatment for the described diseases are shown in Table 1 and Table 2; Figure 2.

## 9. Conclusions

All the above-mentioned rare hematological diseases have a genetic cause. Some of them are described in the literature as diseases caused by mutations in a single gene, like mastocytosis, thrombotic thrombocytopenic purpura, Gaucher disease, and paroxysmal nocturnal hemoglobinuria. In others, like histiocytosis, hemophagocytic lymphohistiocytosis, and Castleman disease, several mutation variants are possible. Performing genetic testing is not obligatory to make a diagnosis, and it serves more often as confirmation of a diagnosis. Unfortunately, genetic diagnosis of rare diseases sometimes takes place at the end of the diagnostic process. Delays in the diagnostic process might translate into unfavorable treatment results.

An explanation of the molecular basis of rare diseases in hematology leads to a better understanding of the pathogenesis and prognosis of the disease and may allow for the development of new molecularly targeted therapies. There is a need for further molecular investigations to discover other possible defects in genes that are responsible for rare diseases.

## Figures and Tables

**Figure 1 cells-12-00449-f001:**
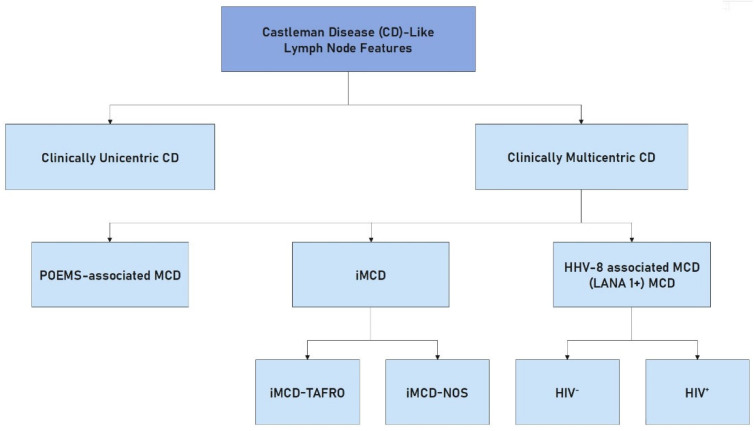
Classification of Castleman disease.

**Figure 2 cells-12-00449-f002:**
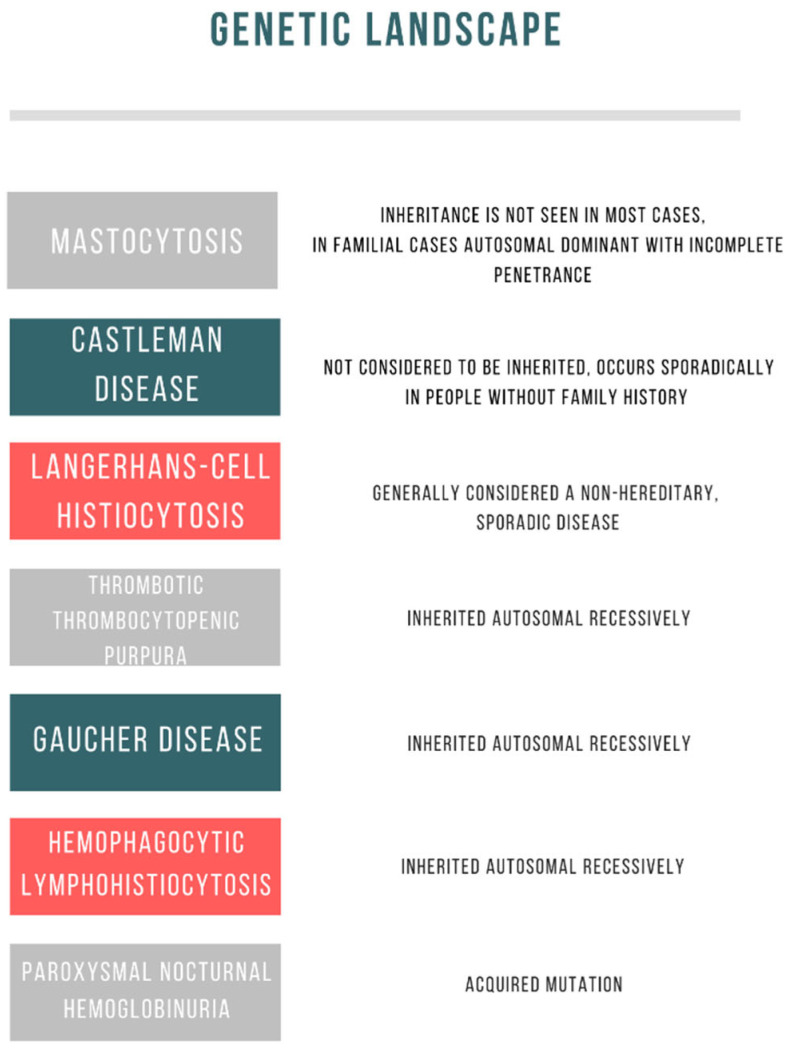
Genetic causes of rare diseases in pediatric hematology.

**Table 1 cells-12-00449-t001:** Rare diseases in pediatric hematology and their genetic causes.

Diseases	Gene
Mastocytosis	*KIT*
Castleman disease	*ETS1, PTPN6, TGFBR2, DNMT3A, PDGFRB*
Histiocytosis	*BRAF, MAP2K1, MAP3K1, ARAF, ERBB3, NRAS, KRAS, PICK1, PIK3R2, PIK3CA*
Trombotic trombocytopenic purpura	*ADAMTS13*
Gaucher disease	*GBA1*
Hemophagocytic lymphohistiocytosis	*HPLH1, PRF1, UNC13D, STX11, STXBP2, SH2D1A, BIRC4, ITK, CD27, MAGT1, LYST, AP3B1, RAB27A*
Paroxysmal nocturnal hemoglobinuria	*PIG-A*

**Table 2 cells-12-00449-t002:** Examples of methods of treatment of rare diseases in hematology.

Diseases	Novel Treatment
Mastocytosis	leukotriene antagonists, H1 and H2 antihistamines, cromolyn sodium, corticosteroids, methoxypsoralen therapy with long-wave psoralen plus ultraviolet A, midostaurin, imatinib, nilotinib, dasatinib, masitinib, avapritinib, ripretinib, cladribine
Castleman disease	corticosteroids, rituximab, thalidomide, lenalidomide, bortezomib, cyclosporine, sirolimus, interferon, antiretroviral therapy
Langerhans-cell histiocytosis	vemurafenib, dabrafenib, imatinib
Trombotic trombocytopenic purpura	therapeutic plasma exchange with fresh frozen plasma replacement, corticosteroids, cyclosporine A, mycophenolate mofetil, rituximab, bortezomib, caplacizumab
Gaucher disease	enzyme replacement therapy, imiglucerase, velaglucerase, taliglucerase, substrate reduction therapy, miglustat, eliglustat
Hemophagocytic lymphohistiocytosis	corticosteroids, etoposide, cyclosporin, emapalumab, anakinra, ruxolitinib, tofacitnib, baricitinib, itacitinib
Paroxysmal nocturnal hemoglobinuria	anti-thrombosis prophylaxis, blood transfusion, allogeneic bone marrow transplantation, eculizumab, pegcetacoplan

## Data Availability

No new data were created or analyzed in this study. Data sharing is not applicable to this article.

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
