# Peer review of "Novel Molecular Therapies and Genetic Landscape in Selected Rare Diseases with Hematologic Manifestations: A Review of the Literature"

_cells, 2023, doi:10.3390/cells12030449_

Round 1

Reviewer 1 Report

The manuscript presents a review of modern scientific literature on the features of molecular pathogenesis in mastocytosis, Castleman's disease, Langerhans cell histiocytosis, thrombotic thrombocytopenic purpura, Gaucher disease, hemophagocytic lymphohistiocytosis, and paroxysmal nocturnal hemoglobinuria. The article should certainly be of interest to readers of the "Cells" journal. However the title of the manuscript (rare diseases with hematologic manifestations) sound more general than the data discussed. There are plenty of other rare conditions that are associated with hematologic manifestations in addition to those described in the article. Like Wilson disease or Gilbert's syndrome among inherited. A lot of rare acquired clonal lympho- or myeloproliferative disorders are also have hematologic manifestations by definition. The article should definitely benefis if authors could give a more specific title to the manuscript or discuss in more detail why exactly the listed diseases were selected for the review. Other minor points or typos are:

Line 118 “for pa-tients” - typo

Line 133 “a single-center Castleman disease (UCD)” – UCD could rather be reffered as Unicentric Castleman disease or Localized Castleman disease

Line 184: “by the whole sequencing. exome.” - the word order is looks to be mixed up

Line 244: “SS-bone LCH” abbreviation should be explained

Line 304: “in a randomized controlled study phase 2” - the word order might be improved

Line 372: “gene loci (… … ), which codes proteins” – better use “code” for plural loci

Line 409: “erythrocytes hemolyze” – perhaps this is a tautology

Author Response

Dear Reviewer,

Thank you very much for Your comments. We have been able to incorporate changes to reflect the suggestions provided. We have highlighted the changes within the manuscript. Here is a point-by-point response to the reviewers’ comments and concerns:

Reviewer 1:

The article should definitely benefit if authors could give a more specific title to the manuscript or discuss in more detail why exactly the listed diseases were selected for the review.

It has been slightly changed because the second reviewer pointed out that in the title there is “excellent naming”. We highlighted that those diseases were selected, and we did not analyze every rare disease with hematologic manifestations.

Line 118 “for pa-tients” – typo

It has been changed.

Line 133 “a single-center Castleman disease (UCD)” – UCD could rather be reffered as Unicentric Castleman disease or Localized Castleman disease

It has been changed.

Line 184: “by the whole sequencing. exome.” - the word order is looks to be mixed up

Thank You for pointing this out. We changed it to “whole exome sequencing”

Line 244: “SS-bone LCH” abbreviation should be explained

We explained the abbreviation.

Line 304: “in a randomized controlled study phase 2” - the word order might be improved

We agree with this point. We changed it to “the second phase of randomized controlled trial”.

Line 372: “gene loci (… … ), which codes proteins” – better use “code” for plural loci

It has been changed.

Line 409: “erythrocytes hemolyze” – perhaps this is a tautology

Thank You for pointing this out. We removed the unnecessary word “erythrocytes”.

Reviewer 2 Report

I reviewed the manuscript written by Dr. Gabriela RÄ™ka et al.; entitled “Novel molecular therapies and genetic landscape in rare diseases with hematologic manifestations – a review of the literature.” This review article provided me completely new aspect for hematological rare diseases. They called those rare hematological diseases as “hematological manifestations”, which is excellent naming. When encountering such hematological disorders, we consider characterizing a genetic feature which can explain the hematological abnormal condition. This review article highlighted an essence handling rare hematological manifestations by genetic aspect. I pointed out some minor issues. I would be glad if you could correspond to my offer.

1. You mentioned target therapy for IL-6 for Castleman disease (page 4, line 199). It would be tocilizumab. Please refer the drug name.

2. Complement inhibitor eculizumab (page 9, line 429) and ravulizumab are available for treatment of paroxysmal nocturnal hemoglobinuria. Please add ravulizumab with eculizumab.

Author Response

Dear Reviewer,

Thank You very much for Your comments. 

  1. You mentioned target therapy for IL-6 for Castleman disease (page 4, line 199). It would be tocilizumab. Please refer the drug name.

It has been added.

  1. Complement inhibitor eculizumab (page 9, line 429) and ravulizumab are available for the treatment of paroxysmal nocturnal hemoglobinuria. Please add ravulizumab with eculizumab.

We found this remark very helpful. We found a new reference and added one sentence concerning ravulizumab.

Best regards

Monika Lejman